# Agnostic Membership Inference Attack on Two-Tower Neural Networks for Recommendation Systems

## Abstract

This paper presents an innovative adaptation of the Agnostic Label-Only Membership Inference Attack (ALOA) specifically designed for two-tower neural network (NN) models used in recommendation systems. Unlike traditional membership inference attacks that focus on categorical outputs, our approach targets models that produce continuous vector embeddings. We propose a comprehensive methodology that employs synthetic datasets, shadow model training, and a suite of perturbation techniques to evaluate model robustness using the Maximum Mean Discrepancy (MMD) metric. Experimental results demonstrate that the attack model achieves exceptionally high accuracy and precision in distinguishing whether data is part of the original training dataset $D_o^{\text{train}}$, even without direct access to it. These findings extend the theoretical framework of membership inference attacks to continuous output spaces and highlight vulnerabilities in modern recommendation systems.

## 1 Introduction and Related Work

The rapid advancement and refinement of large language models have significantly streamlined the implementation of deep neural networks (DNNs), facilitating their widespread adoption across diverse domains. This progress has attracted substantial investments, resulting in an increasing number of companies and deep learning (DL) applications becoming integral to our daily lives. DL models are now omnipresent, influencing activities from browsing websites and watching videos to online shopping and locating the best local restaurants. While DL models, often branded as "AI" features, enhance convenience and efficiency, they also introduce substantial risks related to privacy leakage and potential poisoning attacks that can disrupt model functionality. Sophisticated attackers can exploit DL models trained on personal data to expose sensitive information, such as individuals' relationships and residential locations, to unauthorized parties.

Among the various types of attacks targeting DL models, a few stand out due to their potential impact. The Label-Only Membership Inference Attack (MIA), introduced by Shokri et al. (2017), is a prominent example. MIA enables attackers to determine whether specific data points were included in the training dataset of a target DL model using only the output labels. When combined with Generative Adversarial Networks (GANs), this method escalates the risk by enabling the reconstruction of sensitive data, particularly when models are trained on confidential information belonging to individuals or organizations Fredrikson et al. (2015). Additionally, the industry's shift toward data trading by some governments and corporations exacerbates the risk of digital property theft, making attacks like MIA increasingly concerning.

Building upon these foundational attacks, Monreale et al. (2023) introduced the Agnostic Label-Only Attack (ALOA). ALOA generalizes MIA by removing dependence on the original dataset's distribution, thereby broadening the scope and applicability of membership inference attacks. However, existing research on ALOA predominantly focuses on categorical models with discrete labels, leaving a significant gap in understanding its implications on more complex models that produce continuous outputs, such as those generating vector embeddings.

Privacy has consistently been a critical concern across various fields, receiving significant attention due to the increasing reliance on data-driven technologies. Privacy information leakage can occur through multiple avenues, including direct data access or indirectly via deep learning (DL) models. Since DL models are

trained on datasets and subsequently converge into mathematical representations, they inherently encapsulate information from the training data. Consequently, if an adversary intentionally queries the model, it becomes feasible to exploit and extract sensitive information or at least gain substantial knowledge about the original dataset.

In the realm of data privacy, the initial step involves employing privacy risk assessment methodologies to evaluate the privacy risks associated with users within a dataset. Based on these assessments, appropriate privacy protection techniques are developed to safeguard the data or the DL models themselves. Common privacy protection strategies in DL include randomization, differential privacy, and k-anonymity; see Dwork & Roth (2014), Sweeney (2002) and Duchi et al. (2018). Additionally, emerging research in data decomposition, such as the concept of data elements, aims to break down data into smaller components. This decomposition not only maintains the efficacy of DL models trained on the transformed data but also significantly reduces the risk of data leakage, sometimes achieving zero risk; see Cummings & Zarsky (2021).

Focusing on privacy risk assessment within DL models, the PRUDEnce framework, proposed by Pratesi et al., facilitates a systematic evaluation of empirical privacy risks concerning specific privacy attacks on data; see Pratesi et al. (2020). PRUDEnce simulates an adversary equipped with the knowledge necessary to maximize the privacy risk for each individual in the dataset by generating all possible background knowledge that the adversary might possess and assessing the associated risks under the most adverse conditions.

Over the past few years, several methodologies have been developed to target DL models directly when datasets have privacy risks. Notable among these are:

- **Membership Inference Attack (MIA)**: Introduced by Shokri et al., MIA aims to determine whether a particular data point was part of the training dataset of a classification model by leveraging the model's outputs to infer membership, posing significant privacy risks, especially for sensitive or proprietary datasets; see Shokri et al. (2017).

- **Reconstruction Attack**: Proposed by Fredrikson et al., this attack seeks to reconstruct one or more training samples along with their corresponding labels by exploiting the model's parameters and outputs, thereby undermining data privacy; see Fredrikson et al. (2015).

- **Property Inference Attack**: This attack extracts unintended and non-specific information learned by the model, such as the overall distribution of the training dataset. Unlike MIA, which focuses on individual data points, property inference targets aggregate properties, potentially revealing sensitive characteristics of the dataset; see Yeom et al. (2018).

Building upon these foundational attacks, Choquette-Choo et al. introduced the **LABELONLY attack**, which extends MIA by operating without access to the probability vector outputs of the model. This attack demonstrates the ability to infer membership using only the final labels, thereby lowering the barrier for successful attacks in scenarios where full probability information is unavailable; see Choquette-Choo et al. (2019).

Further advancing this line of research, Anna Monreale et al. proposed the **Agnostic Label-Only Membership Inference Attack (ALOA)** in Monreale et al. (2023). ALOA enhances the LABELONLY attack by eliminating the requirement for knowledge about the training dataset's distribution, allowing broader applicability and effectiveness in diverse settings. Monreale et al. demonstrated that ALOA achieves desirable performance metrics in privacy attacks on categorical models, marking a significant step forward in membership inference methodologies; see Monreale et al. (2023).

## 1.1 Our contributions

Despite these advancements, existing research on ALOA has predominantly concentrated on categorical models with discrete labels, leaving a substantial gap in understanding its implications on more complex models that produce continuous outputs. For instance, in recommendation systems, the models output embeddings in a continuous vector field, representing users and items, which are then utilized by ranking

This paper addresses this gap by pioneering the application of ALOA to **two-tower neural network (NN) models**, which generate outputs in continuous vector fields rather than discrete labels. Two-tower NN models are widely employed in recommendation systems and information retrieval tasks; see Kato et al. (2019) and He et al. (2017), where they process information from two distinct towers—users and items—and output embeddings for each. These embeddings are then utilized by ranking systems to generate personalized recommendations based on relevance and geometric patterns. Unlike previous studies that concentrate on categorical outputs, our approach tackles the unique challenges posed by continuous output spaces, making it a novel contribution to the field.

The primary contribution of this research lies in developing and validating an enhanced attack methodology that effectively infers membership and reconstructs sensitive information from vector-based embeddings in two-tower NN models. This advancement is significant because two-tower NN models are prevalent in recommendation systems, where the integrity and privacy of user and item embeddings are crucial. Furthermore, our research operates under the realistic assumption that the attacker has access only to input features obtainable from the target model, without prior knowledge of the original dataset's distribution. This scenario mirrors real-world conditions more closely than many existing studies, thereby increasing the practical relevance and applicability of our findings.

By demonstrating that ALOA can be successfully adapted to complex embedding-based models, this paper not only extends the theoretical framework of membership inference attacks but also provides empirical evidence of its feasibility in practical settings. This highlights the urgent need for enhanced security measures to protect sensitive data in sophisticated DL models.

## 2 Background

Before proceeding with the details of validating ALOA efficiency in complex multilayer neural networks, we introduce some basic notions that will be used or useful for understanding the approach and modifications used in our research.

### 2.1 Classifier

Suppose that $x = (a, v)$ is a feature variable in which $a$ is the feature name and $v$ is the corresponding $m$-dimensional feature value. Suppose $y$ is the class label of $x$. The feature name $a$ may be either continuous or categorical. For instance, in our real data, $a \in \{$id, zip code, occupation, gender, age bucket$\}$. In practice, we observe $n$ feature variables $x_1 = (a_1, v_1), \ldots, x_n = (a_n, v_n)$ and their corresponding class labels $y_1, \ldots, y_n$.

In classic literature, one aims to construct a *classifier* $b : x \to y$ that sends an input instance $x$ from a feature space to a class label $y$. The classifier produces a categorical output $y$, expressed as $y = b(x)$. The Agnostic Label-Only Attack (ALOA) assumes that the target classifier $b$ is accessible and can be queried without restrictions. In practice, the classifier $b$ is trained on a dataset $D_o^{\text{train}}$. However, classifiers are limited to producing categorical outputs. In more complex scenarios, it is often desirable for the model to generate continuous scalar or vector outputs. In the following section, we introduce an alternative approach based on the Two-Tower Neural Network.

### 2.2 Two-Tower Neural Network Models

Two-Tower Neural Network (TTNN) models Chapelle et al. (2011) are a popular architecture in recommendation systems, especially suited for large-scale retrieval tasks. The core idea behind two-tower models is to learn separate embedding functions for users and items, such that the resulting embeddings can be efficiently compared—typically using dot product or cosine similarity—to compute relevance scores.

A TTNN is a black-box model that consists of two parallel subnetworks:

- **User Tower** $f_u : \mathcal{U} \to \mathbb{R}^d$

- **Item Tower** $f_v : \mathcal{V} \to \mathbb{R}^d$

where $\mathcal{U}$ and $\mathcal{V}$ represent the spaces of user and item features respectively, and $d$ is the dimensionality of the embedding space.

Each tower processes its respective input feature vector:

- User features $u = (u_1, u_2, \ldots, u_m) \in \mathcal{U}$ may include user ID, gender, zip code, bucketized age, and occupation.

- Item features $v = (v_1, v_2, \ldots, v_n) \in \mathcal{V}$ may include movie ID, genre, title, and release year.

The towers map these raw input vectors $u, v$ to dense vector embeddings:

$$\mathbf{u} = f_u(u) \in \mathbb{R}^d, \quad \mathbf{v} = f_v(v) \in \mathbb{R}^d.$$

The output of TTNN is represented as $s(u, v)$ which is the relevance score between user $u$ and item $v$ that can be computed as either the dot product

$$s(u, v) = \langle \mathbf{u}, \mathbf{v} \rangle = \sum_{i=1}^{d} \mathbf{u}_i \mathbf{v}_i,$$

or cosine similarity

$$s(u, v) = \frac{\langle \mathbf{u}, \mathbf{v} \rangle}{\|\mathbf{u}\| \, \|\mathbf{v}\|}.$$

During training, both towers are optimized to maximize the score for observed (user, item) pairs and minimize it for negatives, often via a contrastive loss such as:

$$\mathcal{L} = -\log \frac{\exp\big(s(u, v^+)\big)}{\exp\big(s(u, v^+)\big) + \sum_{v^- \in \mathcal{N}} \exp\big(s(u, v^-)\big)},$$

where $v^+$ is a positive item for user $u$ and $\mathcal{N}$ is a set of negative samples.

At inference time, item embeddings can be precomputed and stored. Given a new user embedding $\mathbf{u}$, similarity scores against the item index allow fast top-$k$ retrieval using approximate nearest neighbor search. This decoupled design is key to large-scale recommendation efficiency.

## 2.3 Black-Box Model

In existing literature, a *black-box setting* refers to an attack scenario where neither the classifier's architecture nor the training dataset is known. In our work, we extend this concept to neural networks. Hereafter, the term "Black-Box model" refers to the target model being attacked without any internal knowledge of its structure or training data.

## 2.4 Shadow Model

*Shadow models* are deep learning models trained to approximate the behavior of a Black-Box model. They play a crucial role in conducting Membership Inference Attacks (MIA) by generating datasets on which attack models $\{A(\cdot)\}$ are trained. A set of shadow models $\{s(\cdot)\}$ is trained using a synthetic dataset $\widetilde{X}$, which is constructed to resemble the structure of the Black-Box model's input. The corresponding outputs predicted by the Black-Box model are denoted as $\widetilde{Y} = b(\widetilde{X})$. The shadow models are subsequently trained on the dataset: $D_s^{\text{train}} = \{(\widetilde{x}_1, \widetilde{y}_1), (\widetilde{x}_2, \widetilde{y}_2), (\widetilde{x}_3, \widetilde{y}_3), \ldots, (\widetilde{x}_n, \widetilde{y}_n)\}$, with the objective of replicating the behavior of $b$. Notably, $D_s^{\text{train}}$ is disjoint from the original training dataset $D_o^{\text{train}}$ due to the inherent lack of access to $D_o^{\text{train}}$. The purpose of training a shadow model is to approximate and ultimately reconstruct $D_o^{\text{train}}$. Chapelle et al. (2011) has demonstrated that training a single shadow model to mimic the entire output

space can achieve performance comparable to training separate shadow models for each category. Therefore, in this study, we employ a single shadow model.

Shadow models serve as a mechanism to infer information about the Black-Box model. As established in Theorem 1, for any $x \in X$ such that $b(x) = y$, there exists a function $s_i(\cdot)$ within the set of shadow models $\{s_1(\cdot), s_2(\cdot), s_3(\cdot), \ldots, s_m(\cdot)\}$ such that $s_i(x) = y$.

**Theorem 1** (Representation Theorem for Shadow Models, adapted from Ye et al. (2022)). *For all $x \in X$ such that $b(x) = y$, there exists a function $s_i(\cdot)$ in the set of shadow models $\{s_1(\cdot), s_2(\cdot), s_3(\cdot), \ldots, s_m(\cdot)\}$ such that $s_i(x) = y$.*

As a consequence, the probability vectors predicted by $\{s_i(\cdot)\}$ are expected to closely approximate those produced by $b$. This similarity can be exploited to infer whether a specific data point belongs to the original training dataset.

### 2.5 Membership Inference Attack (MIA)

Shokri et al. (2017) demonstrated that a classifier $b$ trained on a dataset $D_{\text{train}}^b$ retains relationships between its outputs and the original data. To perform an attack on $b$, a model $A(\cdot)$ is trained to identify whether a data point was included in $D_{\text{train}}^b$. A synthetic dataset $\bar{D}_s^{\text{train}}$ is created and queried through $b$, generating outputs $O_s$. The attack model $A(\cdot)$ is a binary classifier that predicts *IN* if the input data belongs to $D_{\text{train}}^b$, or *OUT* otherwise. $A(\cdot)$ is trained on a dataset $D_s^{\text{train}} = \{(x_i, y_i)\}_s$, where each $x_i$ includes the label predicted by $b$ and the probability vector $\bar{y}_i$ of length $L$. The elements of $\bar{y}_i$ are obtained by querying one of the shadow models $\{s(\cdot)\}$ trained on $\bar{D}_s^{\text{train}} \cup O_s$ to mimic $b$'s behavior for specific categories. Since $\{s(\cdot)\}$ outputs vectors of dimension $L$, there are $L$ models in $\{A(\cdot)\}$, each corresponding to one category. The model $A(\cdot)$ computes the probability $\Pr\{(x, y) \in D_b^{\text{train}}\}$ that the input data is part of $D_{\text{train}}^b$ based on the distribution in $\bar{y}^i$.

### 2.6 Label-Only Membership Inference Attack (LabelOnly)

Choquette-Choo et al. (2019) introduced the LabelOnly attack, which relaxes the requirements of MIA by operating without access to the probability vector outputs of the model. Unlike MIA, which requires probability vectors, LabelOnly relies solely on hard labels predicted by $b$. It utilizes the model's robustness to perturbations to infer membership. Since deep learning models often incorporate perturbations during training to prevent overfitting, their outputs demonstrate higher resilience to minor changes for training data. The LabelOnly model $A_{LO}(\cdot)$ employs shadow models to infer membership. A shadow model $s_{LO}(\cdot)$ is trained on $D_s^{\text{train}} = \{(x^i, y^i)\}$ with a structure and distribution similar to $D_o^{\text{train}}$. The predicted labels by $b$ are recorded in $D_s^{\text{train}}$. The model $A_{LO}(\cdot)$ is trained by perturbing $D_s^{\text{train}}$ to obtain predicted labels $\widetilde{y}^i = s(\widetilde{D}_s^{\text{train}})$, where $\widetilde{D}_s^{\text{train}}$ is the perturbed dataset. The robustness score for each data point $x^i$ is computed as the percentage of labels that remain unchanged after perturbation. An iterative thresholding procedure classifies records as *IN* or *OUT* based on their robustness scores, with higher scores indicating a higher probability of being in $D_o^{\text{train}}$.

### 2.7 Agnostic Label-Only Membership Inference Attack (ALOA)

Monreale et al. (2023) advanced the LabelOnly attack by removing the necessity of knowledge about the original data distribution. Instead, they revised the function for computing the robustness score. The revised robustness score is defined as:

$$\text{rScore}_{x_s^i}(N_{x_s^i}) = \begin{cases} 0 & \text{, if } s(x_s^i) \neq b(x_s^i); \\ \frac{1}{|N_{x_s^i}|} \sum_{x' \in N_{x_s^i}} F(s(x'), s(x_s^i)) & \text{, otherwise,} \end{cases}$$

where $N_{x_s^i}$ denotes the set of entries $x'$ generated by introducing noise around $x_s^i$. The function $F(s(x'), s(x_s^i))$ returns a value close to 0 if $s(x') \neq s(x_s^i)$ and close to 1 otherwise. Thus, the robustness score ranges between 0 and 1.

Instead of requiring knowledge of the original dataset $D_o^{\text{train}}$, ALOA generates a synthetic dataset $x_s^{\text{train}}$ with the same structure as the Black-Box model's input. This dataset is queried through both the shadow model and the target model. The set

$$D_s^{\text{train}} = \{x_s^i \in x_s^{\text{train}} \mid s(x_s^i) = b(x_s^i)\}$$

is constructed. Noise is then introduced to entries in $D_s^{\text{train}}$ to compute robustness scores. Subsequently, ALOA proceeds similarly to the LabelOnly approach without requiring distribution knowledge.

Mathematically, our approach leverages concepts from the mathematics of deep learning by Vidal (2018) to analyze the stability and robustness properties of neural network classifiers under perturbations, providing theoretical support for the ALOA methodology.

## 3  Proposed Approach

To implement ALOA on Two-Tower NN models, we follow a structured methodology comprising model setup, shadow model training, perturbation application, robustness scoring, and attack model training. This section provides a comprehensive overview of each step, illustrating how they collectively contribute to the successful execution of the ALOA attack.

Given the Black-Box model $b$, we extracted the input structure required by the model. A synthetic dataset

$$X_{\text{user}}^{\text{train}} \cup X_{\text{movie}}^{\text{train}} = X_s^{\text{train}}$$

with an identical structure was generated. Focusing on user data, which is typically more informative and valuable, we employed two distinct approaches to train the shadow model, thereby influencing the overall attack accuracy:

1. **Complete Approach**: Utilizes the entire synthetic dataset $X_s^{\text{train}} = X_{\text{user}}^{\text{train}} \cup X_{\text{movie}}^{\text{train}}$, where both user and movie data are synthetically generated to mimic the distribution of the original data.

2. **Pseudo Approach**: Uses only the user portion $X_s^{\text{train}} = X_{\text{user}}^{\text{train}}$, and assigns a fixed pseudo entry for movie data (e.g., a constant vector of all 1's) for every record.

Although the input $X_s^{\text{train}}$ includes both user and movie features, we apply perturbations exclusively to the user input for two key reasons. First, our goal is to simulate realistic adversarial scenarios, where malicious actors are more likely to target user information due to its higher sensitivity and greater privacy implications. Leaks involving user data—such as age, location, or occupation—pose a more significant threat than item data, making user-targeted attacks more representative of real-world risks. Second, although we initially considered investigating leakage from movie features as well, the high accuracy achieved using user perturbations alone already demonstrates the effectiveness of our method in identifying membership. This suggests that even partial input manipulation is sufficient to expose vulnerabilities. Lastly, the rationale behind including both the complete and pseudo approaches is to assess whether the inclusion or exclusion of movie data in shadow model training affects the performance of the attack. This comparative analysis will be discussed in detail in later sections following the presentation of experimental results.

For each $x_i \in X_s^{\text{train}}$, the corresponding labels $Y_s^{\text{train}} = \{y_i \mid y_i = b(x_i)\}$ were generated by querying the Black-Box model $b$. The shadow model $s$ was then trained on the dataset $D_s^{\text{train}} = X_s^{\text{train}} \cup Y_s^{\text{train}}$ to accurately replicate $b$'s prediction behavior.

To systematically compute robustness scores, we introduce a series of perturbations to the user data in $D_s^{\text{train}}$. These perturbations are designed to emulate real-world noise and inaccuracies commonly encountered in user data. Based on variations in different input features—such as ID, Gender, Zip Code, Bucketized Age, and Occupation—we define the following perturbation methods:

1. **User ID Perturbation**
   - **Method**: Add or subtract a random integer within the range $[-10, 10)$ to the original user ID.

- **Outcome**: A new dataset with slightly altered user IDs, maintaining valid values within acceptable bounds.

2. **Gender Flip Perturbation**

   - **Method**: Invert the binary gender value; if coded as 0 (e.g., Female), change to 1 (e.g., Male), and vice versa.
   - **Outcome**: A dataset where each user's gender has been inverted, with all other attributes remaining unchanged.

3. **Zip Code Perturbation**

   - **Method**: Add or subtract a random integer within the range $[-50, 50)$ to each user zip code.
   - **Outcome**: A dataset with slightly varied zip codes that remain valid and within the model's expected embedding range.

4. **User Bucketized Age Shift Perturbation**

   - **Method**: Shift each user's bucketized age to an adjacent age bucket. For middle buckets, randomly decide to shift up or down one bucket. For boundary buckets, shift to the only available adjacent bucket.
   - **Outcome**: A dataset where each user's age bucket is shifted to a neighboring bucket, simulating slight age misclassifications or boundary adjustments.

5. **Occupation Change Perturbation**

   - **Method**: Randomly assign a different occupation label to each user from the set of valid occupation labels, excluding the current one. Ensure that the new label remains within the model's expected range.
   - **Outcome**: A dataset where each user has a randomly altered occupation label within valid bounds, simulating misclassification or noise in occupation data.

Following the application of perturbations, we ensured data integrity and model compatibility by verifying that each altered feature remained within the expected value ranges. This validation prevents runtime errors during model inference, such as out-of-bounds errors in embedding layers. Since only one target feature is modified at a time while all other features remain unchanged, any observed changes in the model's behavior can be solely attributed to the perturbed feature.

Each perturbed dataset was saved in dedicated directories corresponding to the type of perturbation (e.g., `user_id_perturbation`, `gender_flip`, etc.) to facilitate systematic testing and comparison across different perturbation types. Identical perturbations were also applied to $D_o^{\text{train}}$ for baseline comparisons.

To quantify the impact of each perturbation, we computed the Maximum Mean Discrepancy (MMD) between the embeddings obtained from the original and perturbed datasets. MMD is a statistical measure used to compare the difference between two distributions. Given two sets of embeddings, $\mathcal{X} = \{\mathbf{u}_i\}_{i=1}^n$ from the original data and $\mathcal{Y} = \{\mathbf{v}_j\}_{j=1}^m$ from the perturbed data, the squared MMD is computed as:

$$\text{MMD}^2(\mathcal{X}, \mathcal{Y}) = \frac{1}{n^2} \sum_{i=1}^n \sum_{j=1}^n k(\mathbf{u}_i, \mathbf{u}_j) + \frac{1}{m^2} \sum_{i=1}^m \sum_{j=1}^m k(\mathbf{v}_i, \mathbf{v}_j) - \frac{2}{nm} \sum_{i=1}^n \sum_{j=1}^m k(\mathbf{u}_i, \mathbf{v}_j),$$

where $k(\cdot, \cdot)$ is a kernel function (e.g., the Gaussian kernel defined as $k(\mathbf{u}, \mathbf{v}) = \exp\left(-\frac{\|\mathbf{u}-\mathbf{v}\|^2}{2\sigma^2}\right)$). A larger MMD value indicates a larger discrepancy between the two distributions, reflecting a greater impact of the perturbation on the model's output.

In order to systematically apply these perturbations and facilitate reproducibility, we define a function called `perturbation` that performs the specified perturbations on the dataset. Below is Algorithm 1, which outlines the detailed procedure for generating perturbed datasets.

---

**Algorithm 1** Perturbation Function for ALOA

---

**Require:** $D$ - Original user dataset
**Require:** $perturb\_type$ - Type of perturbation to apply ({User ID Perturbation, Gender Flip, Zip Code Perturbation, User Bucketized Age Shift, Occupation Change})
**Require:** $params$ - Additional parameters required for the perturbation (e.g., range limits)
**Ensure:** $D_{perturbed}$ - Perturbed user dataset
1: **if** $perturb\_type =$ "User ID Perturbation" **then**
2:      Convert `user_id` to numeric values
3:      Generate random integers $r_i \in [-10, 10)$ for each user
4:      Update `user_id`: $user\_id \leftarrow user\_id + r_i$
5:      Clip `user_id` within range $[0, 6040]$
6: **else if** $perturb\_type =$ "Gender Flip" **then**
7:      Confirm that `user_gender` is binary (0 or 1)
8:      Flip `user_gender`: $user\_gender \leftarrow 1 - user\_gender$
9: **else if** $perturb\_type =$ "Zip Code Perturbation" **then**
10:      Convert `user_zip_code` to numeric values
11:      Generate random integers $r_i \in [-50, 50)$ for each user
12:      Update `user_zip_code`: $user\_zip\_code \leftarrow user\_zip\_code + r_i$
13:      Clip `user_zip_code` to valid range $[30000, 33439]$
14:      Map clipped `user_zip_code` to embedding indices: $user\_zip\_code \leftarrow \max(0, \min(3439, user\_zip\_code - 30000))$
15: **else if** $perturb\_type =$ "User Bucketized Age Shift" **then**
16:      Identify and sort unique `user_bucketized_age` values
17:      Create mapping $age\_map$ from age bucket to index
18:      **for** each user in $D$ **do**
19:          Determine current age bucket index $idx$
20:          **if** $idx = 0$ **then**
21:              Shift to the next higher bucket
22:          **else if** $idx =$ last index **then**
23:              Shift to the next lower bucket
24:          **else**
25:              Randomly choose to shift up or down one bucket
26:          **end if**
27:          Update `user_bucketized_age`
28:      **end for**
29: **else if** $perturb\_type =$ "Occupation Change" **then**
30:      Identify valid occupation labels $O = \{0, 1, \ldots, 20\}$
31:      **for** each user in $D$ **do**
32:          Current occupation $o$
33:          Define possible new occupations: $O' = O \setminus \{o\}$
34:          Randomly select new occupation $o' \in O'$
35:          Assign $o'$ to `user_occupation_label`
36:      **end for**
37: **end if**
38: **return** $D_{perturbed}$

---

# 4 Experiment

This section delineates the experimental setup and results of applying ALOA on Two-Tower NN models. The experiments assess the effectiveness of ALOA in inferring membership based on perturbations in user data.

### 4.1 Black-Box Model Robustness Testing

We utilized the multi-stage two-tower recommender project from GitHub created by Keivanipchihagh (2023), which provides a comprehensive dataset for training and evaluating two-tower neural networks in recommendation systems. The dataset comprises:

- **User Information**: User ID, user zip code, gender, age bracket, and user occupation.

- **Movie Information**: Movie ID, movie name, movie genre, and release year.

- **Rating Information**: Records of user-movie interactions along with the ratings provided.

Using PyTorch, we constructed the Two-Tower NN model and exported it to ONNX format to facilitate efficient querying as a Black-Box model.

We explored two distinct approaches to train the shadow model, addressing the challenge of targeting user data:

1. **Complete Approach**: Utilizes the entire synthetic dataset $X_s^{\text{train}} = X_{\text{user}}^{\text{train}} \cup X_{\text{movie}}^{\text{train}}$, where both user and movie data are synthetically generated to mimic the distribution of the original data.

2. **Pseudo Approach**: Uses only the user portion $X_s^{\text{train}} = X_{\text{user}}^{\text{train}}$, and assigns a fixed pseudo entry for movie data (e.g., a constant vector of all 1's) for every record.

The synthetic data used in shadow model training is constructed by directly querying the ONNX-exported Black-Box model for its input schema. Once the expected input types and shapes are obtained, we generate random entries that conform to these specifications. In cases where the synthetic input falls outside the model's accepted range, the model raises an error and returns the required bounds—allowing us to automatically adjust the data to satisfy input constraints. This process ensures that all synthetic data remains compatible with the Black-Box model, and more importantly, it reinforces the agnostic nature of the attack: the adversary constructs training data solely based on the model's interface, without any access to the original training distribution.

For each $x_i \in X_s^{\text{train}}$, the corresponding labels

$$Y_s^{\text{train}} = \{y_i \mid y_i = b(x_i)\}$$

were generated by querying the Black-Box model $b$. The shadow model $s$ was then trained on the dataset

$$D_s^{\text{train}} = X_s^{\text{train}} \cup Y_s^{\text{train}}$$

to accurately replicate $b$'s prediction behavior.

We applied the five perturbation methods outlined in Section 3 to the shadow training dataset $D_s^{\text{train}}$. Each perturbation was applied individually, and the resulting datasets were stored in separate directories to facilitate systematic evaluation. Following perturbation, we queried the perturbed datasets through the Black-Box model $b$ to obtain the predicted embeddings.

To assess the model's robustness to these perturbations, we computed the Maximum Mean Discrepancy (MMD) between the embeddings of the original and perturbed data:

$$\text{MMD}^2(\mathcal{X}, \mathcal{Y}) = \frac{1}{n^2} \sum_{i=1}^{n} \sum_{j=1}^{n} k(\mathbf{u}_i, \mathbf{u}_j) + \frac{1}{m^2} \sum_{i=1}^{m} \sum_{j=1}^{m} k(\mathbf{v}_i, \mathbf{v}_j) - \frac{2}{nm} \sum_{i=1}^{n} \sum_{j=1}^{m} k(\mathbf{u}_i, \mathbf{v}_j)$$

where $\mathcal{X} = \{\mathbf{u}_i\}$ represents the embeddings from the original data $X$, $\mathcal{Y} = \{\mathbf{v}_i\}$ represents the embeddings from the perturbed data $\widetilde{X}$, and $k(\cdot, \cdot)$ is a kernel function (e.g., the Gaussian kernel). A larger MMD value indicates a larger discrepancy between the two distributions, reflecting a greater impact of the perturbation on the model's output.

The robustness scores for different perturbations are presented in Tables 1, 2, 3, 4, and 5.

Table 1: Robustness Scores for User ID Perturbation

| Dataset | Average | Median | Min | Max |
|---|---|---|---|---|
| Original data with pseudo movie entries | 0.080462 | 0.063381 | 0.000000 | 0.735921 |
| Original data with real movie entries | 0.080462 | 0.063381 | 0.000000 | 0.735921 |
| Synthetic data with pseudo movies | 0.090371 | 0.072155 | 0.000000 | 0.485888 |
| Synthetic data with real movies | 0.090371 | 0.072155 | 0.000000 | 0.485888 |

Table 2: Robustness Scores for Zip Code Perturbation

| Dataset | Average | Median | Min | Max |
|---|---|---|---|---|
| Original data with pseudo movie entries | 0.002234 | 0.000000 | 0.000000 | 0.359842 |
| Original data with real movie entries | 0.002234 | 0.000000 | 0.000000 | 0.359842 |
| Synthetic data with pseudo movies | 0.000000 | 0.000000 | 0.000000 | 0.000000 |
| Synthetic data with real movies | 0.000000 | 0.000000 | 0.000000 | 0.000000 |

Table 3: Robustness Scores for User Occupation Perturbation

| Dataset | Average | Median | Min | Max |
|---|---|---|---|---|
| Original data with pseudo movie entries | 0.033829 | 0.025463 | 0.000000 | 0.240031 |
| Original data with real movie entries | 0.033829 | 0.025463 | 0.000000 | 0.240031 |
| Synthetic data with pseudo movies | 0.035301 | 0.027552 | 0.002022 | 0.223600 |
| Synthetic data with real movies | 0.035301 | 0.027552 | 0.002022 | 0.223600 |

Table 4: Robustness Scores for User Gender Perturbation

| Dataset | Average | Median | Min | Max |
|---|---|---|---|---|
| Original data with pseudo movie entries | 0.063202 | 0.064812 | 0.001643 | 0.128037 |
| Original data with real movie entries | 0.063202 | 0.064812 | 0.001643 | 0.128037 |
| Synthetic data with pseudo movies | 0.069620 | 0.068363 | 0.018424 | 0.122920 |
| Synthetic data with real movies | 0.069620 | 0.068363 | 0.018424 | 0.122920 |

Table 5: Robustness Scores for User Age Bucketized Shift Perturbation

| Dataset | Average | Median | Min | Max |
|---|---|---|---|---|
| Original data with pseudo movie entries | 0.415971 | 0.489709 | 0.143693 | 0.987755 |
| Original data with real movie entries | 0.415971 | 0.489709 | 0.143693 | 0.987755 |
| Synthetic data with pseudo movies | 0.042410 | 0.000000 | 0.000000 | 0.828345 |
| Synthetic data with real movies | 0.042410 | 0.000000 | 0.000000 | 0.828345 |

The robustness scores in Tables 1–5 reveal which user features induce the largest embedding shifts—and thus provide the strongest signals—for our ALOA attack. A small robustness score indicates that the model is robust to that perturbation (the embedding changes very little), while a large robustness score indicates non-robustness (a large embedding shift).

**Age buckets** Shifting the user's age bucket yields the highest average robustness score ($\approx 0.416$) and a maximum near 1.0 on real data. This demonstrates that the model encodes age as a dominant feature: even a one-bucket change almost always perturbs the embedding substantially. For ALOA, age-bucket perturbation therefore provides an exceptionally reliable membership signal.

**User ID** Perturbing the user ID produces a moderate mean robustness score ($\approx 0.08$–$0.09$) with considerable spread (up to 0.74). This shows that the model learns distinct embeddings for different ID slots.

Although less powerful than age shifts, ID perturbation remains a useful channel for membership inference, especially when combined with other features.

**Occupation and gender** Changing occupation yields an average robustness score of $\approx 0.03$–$0.04$, while flipping gender gives $\approx 0.06$–$0.07$. Both have relatively tight distributions, indicating these are meaningful but secondary signals. They still generate consistent embedding differences, making them viable—but lower-precision—perturbations for ALOA.

**Zip code** Zip code perturbation produces near-zero robustness scores on both real and synthetic data, showing the model is effectively invariant to small ZIP changes. As a result, ZIP code provides almost no membership signal and is not suitable for ALOA in this context.

**Conclusions** Overall, the two-tower model is most vulnerable to perturbations in age and user ID, moderately sensitive to occupation and gender, and highly robust to ZIP code.

## 4.2 ALOA

Given that the Black-Box model exhibits robustness to perturbations in the original dataset $D_o^{\text{train}}$, we proceed to configure the ALOA attack. The attack encompasses several key components and processes.

First, we define the original training data as $D_o^{\text{train}} = \{(u_i, v_i, r_i)\}_{i=1}^N$, where $u_i \in U$ represents user features, $v_i \in V$ denotes item features, and $r_i \in R$ signifies ratings. For shadow model training, we prepare two distinct datasets: $D_{s,\text{combined}} = \{(u_j, v_j, r_j)\}_{j=1}^N$ and $D_{s,\text{dummy}} = \{(u_k, v_k, r_k)\}_{k=1}^N$, which correspond to the complete and pseudo approaches, respectively. These shadow models, $s_{\text{combined}}$ and $s_{\text{dummy}}$, are trained on their respective datasets to emulate the Black-Box model's behavior.

Next, we generate perturbed data

$$D_p = \{\texttt{perturbation}(u, \phi) \mid (u, v, r) \in D_s^{\text{train}}, \phi \in \Phi\}$$

where $\Phi = \{\phi_1, \phi_2, \ldots, \phi_5\}$ represents different types of perturbations (User ID, Gender Flip, Zip Code, Age Bucket Shift, Occupation Change). These perturbations simulate realistic variations and potential noise in user data, allowing us to assess the model's robustness and the effectiveness of the ALOA attack.

The ALOA attack procedure involves several sequential steps:

**Feature Extraction**: For each perturbed user $u' \in D_{\text{perturbed}}$, we extract feature vectors by passing them through the shadow models:

$$\mathbf{f}(u') = s(u', \bullet) \in \mathbb{R}^d$$

where $s(u', \bullet)$ denotes the concatenated outputs from both towers of the Two-Tower shadow model, and $d$ is the dimensionality of the feature vector.

**Label Assignment**: Binary labels are assigned based on membership status:

$$y(u') = \begin{cases} 1 & \text{if } u' \text{ is a member (from } D_s^{\text{train}}) \\ 0 & \text{if } u' \text{ is a non-member (not from } D_s^{\text{train}}) \end{cases}$$

**Dataset Construction**: We construct the attack dataset

$$D_{\text{attack}} = \{(\mathbf{f}_i, y_i)\}_{i=1}^{2M},$$

where $\mathbf{f}_i = s(u', \bullet)$ and $y_i \in \{0, 1\}$. Here, data labeled as *IN* corresponds to entries in the shadow model's training set $D_s^{\text{train}}$, while *OUT* corresponds to data outside $D_s^{\text{train}}$.

**Model Training**: The attack model training involves the following steps:

1. **Train-Test Split**: Divide $D_{\text{attack}}$ into training and testing subsets:

$$D_{\text{train}}, D_{\text{test}} = \text{TrainTestSplit}(D_{\text{attack}}, \text{test\_size} = 0.3, \text{random\_state} = 42)$$

2. **Classifier Selection**: Employ a Multi-Layer Perceptron (MLP) classifier $\mathcal{C}$ with hyperparameters $\theta$:

$$\mathcal{C} : \mathbb{R}^d \to \{0, 1\}$$

3. **Hyperparameter Optimization**: Optimize $\theta$ using grid search with cross-validation to maximize the ROC AUC score:

$$\theta^* = \arg\max \ \text{ROC\_AUC}(\mathcal{C}_\theta, D_{\text{train}}, y)$$

4. **Training the Classifier**: Train the classifier with the optimized hyperparameters:

$$\mathcal{C}^* = \mathcal{C}_{\theta^*} \leftarrow \text{Train}(\mathcal{C}_{\theta^*}, D_{\text{train}})$$

**Evaluation Metrics**: We evaluate the performance of $\mathcal{C}^*$ on $D_{\text{test}}$ using the following metrics:

- **Accuracy** ($\mathcal{A}$):

$$\mathcal{A} = \frac{1}{|D_{\text{test}}|} \sum_{i=1}^{|D_{\text{test}}|} \mathbb{I}(\mathcal{C}^*(\mathbf{f}_i) = y_i)$$

- **Precision** ($\mathcal{P}$):

$$\mathcal{P} = \frac{\sum_{i=1}^{|D_{\text{test}}|} \mathbb{I}(\mathcal{C}^*(\mathbf{f}_i) = 1 \wedge y_i = 1)}{\sum_{i=1}^{|D_{\text{test}}|} \mathbb{I}(\mathcal{C}^*(\mathbf{f}_i) = 1)}$$

- **Recall** ($\mathcal{R}$):

$$\mathcal{R} = \frac{\sum_{i=1}^{|D_{\text{test}}|} \mathbb{I}(\mathcal{C}^*(\mathbf{f}_i) = 1 \wedge y_i = 1)}{\sum_{i=1}^{|D_{\text{test}}|} \mathbb{I}(y_i = 1)}$$

- **ROC AUC** ($\mathcal{U}$):

$$\mathcal{U} = \text{ROC\_AUC}(\{p_i\}, \{y_i\})$$

where $p_i = \mathcal{C}^*(\mathbf{f}_i)_1$ denotes the predicted probability for class 1 (member).

**Threshold Determination**: We establish a classification threshold $\tau$ using K-Means clustering on the robustness scores $\{r_i\}$, where $r_i = p_i$:

1. **Clustering**:

$$\text{KMeans}(\{r_i\}, k = 2)$$

2. **Threshold Calculation**:

$$\tau = \frac{c_1 + c_2}{2}, \quad \text{where } c_1 \text{ and } c_2 \text{ are the cluster centers}$$

**Classification Based on Threshold**: We classify each instance based on the determined threshold $\tau$:

$$\mathcal{C}^*(\mathbf{f}_i) = \begin{cases} 1 & \text{if } r_i \geq \tau \\ 0 & \text{otherwise} \end{cases}$$

### 4.3 Algorithm for ALOA Attack

To provide a clear example of the ALOA attack procedure, we present Algorithm 2, which outlines the steps involved in executing the attack.

---

**Algorithm 2** Agnostic Label-Only Membership Inference Attack (ALOA)

---

**Require:** $b$ - Black-Box model
**Require:** $s_{\text{dummy}}, s_{\text{combined}}$ - Shadow models trained on dummy and combined datasets
**Require:** $D_o^{\text{train}}$ - Original training dataset
**Require:** $\Phi$ - Set of perturbation types ({User ID, Gender Flip, Zip Code, Age Bucket Shift, Occupation Change})
**Ensure:** $\mathcal{C}^*$ - Trained attack classifier
**Ensure:** $\tau$ - Classification threshold
1: **for** each shadow model $s \in \{s_{\text{dummy}}, s_{\text{combined}}\}$ **do**
2:      **for** each perturbation type $\phi \in \Phi$ **do**
3:          Generate perturbed dataset $D_p = \{\texttt{perturbation}(u, \phi) \mid u \in D_s^{\text{train}}\}$
4:          Extract features $\mathbf{f}(u') = s(u', \bullet)$ for each $u' \in D_p$
5:          Assign labels $y(u') \leftarrow 1$ if $u' \in D_s^{\text{train}}$, else $y(u') \leftarrow 0$
6:      **end for**
7: **end for**
8: Construct attack dataset $D_{\text{attack}} = \{(\mathbf{f}_i, y_i)\}_{i=1}^{2M}$
9: Split $D_{\text{attack}}$ into $D_{\text{train}}, D_{\text{test}}$ using TrainTestSplit with test_size $= 0.3$ and random_state $= 42$
10: Initialize Multi-Layer Perceptron (MLP) classifier $\mathcal{C}$ with hyperparameters $\theta$
11: Optimize $\theta$ using grid search with cross-validation to maximize ROC AUC on $D_{\text{train}}$
12: Train classifier $\mathcal{C}^*$ with optimized $\theta^*$ on $D_{\text{train}}$
13: Evaluate $\mathcal{C}^*$ on $D_{\text{test}}$ to obtain metrics $\mathcal{A}, \mathcal{P}, \mathcal{R}, \mathcal{U}$
14: Perform K-Means clustering on robustness scores $\{r_i\}$ to determine threshold $\tau$
15: Classify each instance in $D_{\text{test}}$ based on $\tau$
16: **return** $\mathcal{C}^*, \tau$

---

### 4.4 Attack Results

The performance of the ALOA attack was evaluated under two scenarios. First, using the shadow models' training data, the attack performance is summarized in Table 6.

Table 6: Attack Performance Using Shadow Models

| Model Type | Accuracy | Precision | Recall | ROC AUC | Optimal Threshold |
|---|---|---|---|---|---|
| Combined | 0.9787 | 0.9789 | 0.9958 | 0.9946 | 0.8451 |
| Dummy | 0.9507 | 1.0000 | 0.4573 | 0.9406 | 0.1273 |

The results are very promising. Both the accuracy and precision are exceptionally high, demonstrating that the attack model is highly effective at determining whether data is *IN* or *OUT* of the original training dataset $D_o^{\text{train}}$. Initially, we were concerned about potential overfitting; however, the training data for the attack model is significantly imbalanced—with an *IN:OUT* ratio of 4:1 for the complete approach and 1:10 for the pseudo approach. If the attack model were not functioning properly, we would expect an accuracy near 0.8 and 0.91 (akin to random guessing) or a precision that closely mirrors these imbalanced distributions (around 0.8 or 0.1, respectively). In light of this, we are confident that the outstanding performance observed is genuine. For further validation, we applied the attack model to the original training dataset of the Black-Box model by computing the proportion of instances classified as *IN*. This proportion serves as an accuracy measure, and the results are presented in Table 7.

Table 7: Attack Performance on Original Training Data of the Black-Box Model

| Model Type | Accuracy (%) |
|---|---|
| Combined | 99.65 |
| Dummy | 85.95 |

Table 7 demonstrates that the attack model retrieves data from the original training dataset $D_o^{\text{train}}$ of the Black-Box model with exceptional efficiency. Remarkably, the attack model was trained solely on data from the shadow models and had no direct access to $D_o^{\text{train}}$ during the attack process, underscoring its practical applicability. These results confirm that the ALOA method is effective for attacking models that output in a continuous vector space. Moreover, our investigation reveals that for Two-Tower neural networks—or indeed any composite deep learning models—if the synthetic dataset used to train the shadow model lacks sufficient diversity, the attack's efficiency is reduced. This intriguing outcome highlights the importance of comprehensive synthetic data generation and warrants further investigation.

## 5 Conclusion

In this work, we have successfully extended the Agnostic Label-Only Membership Inference Attack (ALOA) to two-tower neural network models, a class of systems that play a critical role in modern recommendation engines. By adapting membership inference techniques to continuous vector embeddings, our approach fills a significant gap in the literature. Using shadow models trained on synthetic datasets and perturbation strategies evaluated via the Maximum Mean Discrepancy (MMD) metric, we demonstrated that our attack model can effectively discern between training and non-training data with exceptionally high accuracy and precision—even when the attacker has no direct access to the original training dataset $D_o^{\text{train}}$.

Beyond the primary technical contributions, our investigation reveals a potentially alarming byproduct of the attack: a clearer understanding of the recommendation model's embedding space. This deeper insight into how content is represented and targeted significantly increases the risk of manipulation. An adversary armed with such detailed knowledge could strategically mass-produce or tailor content to influence the recommendations delivered to specific audiences, thereby exerting control over the information these groups receive. This possibility of orchestrated influence raises serious concerns about the broader societal impact of vulnerabilities in recommendation systems.

Furthermore, our experiments indicate that the efficiency of the attack is strongly influenced by the diversity of the synthetic dataset used to train the shadow model. When this dataset lacks sufficient diversity, the performance of the attack diminishes, emphasizing the need for comprehensive data generation strategies.

Overall, our study underscores the urgent need for robust privacy-preserving mechanisms in recommendation systems. The demonstrated vulnerability to label-only membership inference attacks—coupled with the potential for adversaries to exploit an enhanced understanding of model embeddings for content manipulation—calls for immediate attention from both researchers and practitioners. Future work will focus on developing effective countermeasures and further exploring the interplay between synthetic training data diversity and attack efficacy across various deep learning paradigms.

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
