# OpenReview forum: "Agnostic Membership Inference Attack on Two-Tower Neural Networks for Recommendation Systems"
_TMLR — Rejected by TMLR_

### Review · Reviewer_TAL4 · 2025-05-11

**Summary Of Contributions:**

This paper explores the effectiveness of agnostic label-only (black box, no access to output logits, no access to training distribution) membership inference attacks in recommendation systems that use two-tower neural networks. Through experimental results, the authors show that this attack is very effective (95%+ accuracy), both settings where the the black box model is queryable (combined) and also when it is not (dummy).

**Audience:**

Yes

**Claims And Evidence:**

No

**Requested Changes:**

- (critical): Can a section be added to the background detailing how two-tower NN work in recommendation systems? In particular, since this paper focuses on membership inference paper, can more details be added focusing on the exact input and output (x,y) of two-tower NNs?
- (critical): Can the authors clarify why perturbations are only applied to the user input, if the X includes both user and movie data?
- (critical): How is the synthetic dataset created? Since a key aspect of agnostic attacks is no underlying assumption of the original training dataset, it is key to understand how a synthetic dataset is created for shadow model training
- (critical): I believe a higher robustness score should indicate smaller difference between embedding of original and perturbed data? However, the MMD definition indicates the opposite. Can this be clarified?
- (strengthen): What are the main takeaways of tables 1-5? Since the shadow models are trained on the synthetic data (for agnostic label-only attacks), are the results indicating that the base black box model is less robust against its own training data compared to synthethic data? Why is that?

**Strengths And Weaknesses:**

**Strengths**

- strong results applying ALOA to two-tower NN

**Weaknesses**

- it is unclear how the two-tower NN works, i.e. what is it’s input and output. This makes it more difficult to understand exactly what data we are performing membership inference on (see requested changes)
- missing details in experimental design/results of section 4

---

> ### Author Response · Authors · 2025-05-14
> **Author responses**
>
> Thanks for the excellent reviews. Below, we provide point-by-point responses to the requested changes:
>
> (Critical): Can a section be added to the background detailing how Two-Tower NNs work in recommendation systems? In particular, since this paper focuses on membership inference, can more details be added focusing on the exact input and output (x, y) of Two-Tower NNs?
>
> Response: A subsection has been added to Section 2 that introduces the background of Two-Tower Neural Networks in recommendation systems. It includes a detailed explanation of the exact structure and data types of the input and output.
>
> (Critical): Can the authors clarify why perturbations are only applied to the user input, if the X includes both user and movie data?
>
> Response: First, user information is typically more valuable and thus more likely to be the target of attacks in real-world scenarios. Second, while we initially considered perturbing movie data, we found that attacking only the user features already yields high accuracy, making further perturbation unnecessary. Additional explanation has been added to Section 3 of the revised manuscript.
>
> (Critical): How is the synthetic dataset created? Since a key aspect of agnostic attacks is the absence of assumptions about the original training dataset, it is important to understand how a synthetic dataset is generated for shadow model training.
>
> Response: The synthetic dataset is generated by querying the ONNX model to determine the input types and ranges, then sampling random values within each category accordingly. This approach conforms to the agnostic setting, as it requires no prior knowledge of the original training data. A detailed description has been added to Section 4.1 in the revised manuscript.
>
> (Critical): I believe a higher robustness score should indicate a smaller difference between the embeddings of original and perturbed data. However, the MMD definition indicates the opposite. Can this be clarified?
>
> Response: As stated before Tables 1 through 5, "A larger MMD value indicates a greater discrepancy between the two distributions, reflecting a greater impact of the perturbation on the model's output." We acknowledge, however, that there was an inconsistency in the initial robustness evaluation—cosine similarity was mistakenly used instead of MMD. While we later determined that MMD is a more suitable metric for estimating robustness, the original tables were not updated accordingly. We sincerely apologize for this oversight. The tables have now been revised to reflect the correct MMD-based robustness scores.
>
> (Strengthen): What are the main takeaways of Tables 1–5? Since the shadow models are trained on synthetic data (for agnostic label-only attacks), do the results indicate that the base black-box model is less robust against its own training data compared to synthetic data? Why is that?
>
> Response: The main takeaway from Tables 1–5 is that the black-box model exhibits greater robustness when evaluated on its original training data. Lower robustness scores for features like user ID, occupation, and gender suggest that the model is more resilient to perturbations in those dimensions. The results also show that changes in features such as zip code and age buckets have minimal impact on the model’s output, indicating that these attributes are not strongly encoded in the embeddings.

---

### Review · Reviewer_domC · 2025-06-03

**Summary Of Contributions:**

This paper studies the extension of the Agnostic Label-Only Membership Inference Attack (ALOA) designed explicitly for two-tower neural networks. Based on the continuous output vector instead of the predicted discrete labels in TTNN models, the paper demonstrates a method to conduct ALOA by perturbing the user input. Numerical experiments demonstrate that the proposed attack method successfully identifies the vulnerable attributes and attacks the shadow models trained with either combined data or dummy data.

**Audience:**

Yes

**Claims And Evidence:**

No

**Requested Changes:**

1. Please address the weakness above.

2. The paper should highlight its contribution in the first section. Now it is only hidden in one paragraph in the section.

**Strengths And Weaknesses:**

he Strength:
The paper provides a detailed description of the proposed approach, along with its background.  Detailed information on the attack procedure and evaluation matrices is provided.

Weakness: a few things are unclear to me:
1. It is unclear to me why this paper is a "label-only" attack. The Lebel Only attack requires using only the hard labels instead of the probability vectors. Isn't the paper adopting an agnostic MIA approach?
2. What is the function b(u,v)? From the description of the TTNN, it only has f_u and f_v that have outputs of dimension d.
3. Where is Theorem 1 coming from? I can't find it (or any other form of this theory) in the cited paper.
4. Why, at the beginning, does the paper claim " the Black-Box model exhibits robustness to perturbations in the original dataset"? From the results in Section 4.1, it appears that the Original data has a higher robustness score (MMD), which means it's more vulnerable to perturbations.
5. The conclusion in sec. 4.1 also mismatches with the tables. Gender perturbation (on synthetic data) is more vulnerable than age (based on average score)
6. In sec 4.4, claiming that accuracy near 0.5 indicates the model is not functioning is not accurate. Accuracy less than 0.8 and 0.91 also indicates the attack model is not functioning (akin to outputting a constant guess). Also, the recall of dummy data is slightly low (only 0.4573).

---

> ### Author Response · Authors · 2025-06-06
> **Author responses**
>
> Thanks for the excellent reviews. Below are our point-by-point responses to the requested changes:
>
> 1. It is unclear to me why this paper is a "label-only" attack. The Label Only attack requires using only the hard labels instead of the probability vectors. Isn't the paper adopting an agnostic MIA approach?
>
> Thank you for pointing this out. We agree that the term “label-only” is potentially misleading in our context, as our paper focuses on continuous outputs rather than categorical labels. While we initially adopted the term from the cited ALOA paper, we acknowledge that it does not precisely describe our setting. To avoid confusion, we have removed all instances of “label-only” from the manuscript, including the title, in the current revision.
>
> 2. What is the function b(u,v)? From the description of the TTNN, it only has f\_u and f\_v that have outputs of dimension d.
>
> Thank you for catching this. That was a typo—b(u, v) should have been s(u, v), representing the TTNN output given inputs u and v. In the revised manuscript, we removed all occurrences of b(u, v) and clarified that s(u, v) denotes the TTNN’s embedding output. Please refer to the highlighted revision in Section 2.2.
>
> 3. Where is Theorem 1 coming from? I can't find it (or any other form of this theory) in the cited paper.
>
> You are absolutely right. Theorem 1 was misattributed. It should be credited to the paper *“Label-Only Model Inversion Attack: The Attack That Requires the Least Information”* by Dayong Ye et al. We have corrected the reference accordingly in the updated version.
>
> 4. Why, at the beginning, does the paper claim "the Black-Box model exhibits robustness to perturbations in the original dataset"? From the results in Section 4.1, it appears that the Original data has a higher robustness score (MMD), which means it's more vulnerable to perturbations.
>
> We agree that a higher MMD implies greater vulnerability. However, in our results, the original dataset only shows significantly higher scores for age and ZIP code. The model exhibits strong robustness to ZIP perturbations, which supports our conclusion that ZIP contributes little to the embeddings. Feature influence in real-world models is often context-dependent. While the high age sensitivity may stem from the perturbation method, our attack accuracy remains unaffected, reinforcing the method’s robustness.
>
> 5. The conclusion in sec. 4.1 also mismatches with the tables. Gender perturbation (on synthetic data) is more vulnerable than age (based on average score).
>
> We believe there is no mismatch. In Section 4.1, we concluded that the model is most vulnerable to age and user ID perturbations, moderately sensitive to occupation and gender, and robust to ZIP code. This aligns with the robustness score tables: age and user ID consistently yield higher scores than other features. Gender and ZIP have noticeably lower scores, supporting our stated ranking.
>
> 6. In sec 4.4, claiming that accuracy near 0.5 indicates the model is not functioning is not accurate. Accuracy less than 0.8 and 0.91 also indicates the attack model is not functioning (akin to outputting a constant guess). Also, the recall of dummy data is slightly low (only 0.4573).
>
> Thank you. We agree that 0.5 is a low benchmark and that 0.8 or 0.91 is more appropriate. We’ve updated the text to reflect a 0.91 threshold. Still, our reported training accuracies—0.9787 (combined) and 0.9507 (dummy) in Table 6—remain above these thresholds. Moreover, when applied to the original training data, the model achieves 0.9965 accuracy for the combined setup (Table 7). Thus, the conclusions remain valid. Regarding the dummy model’s recall (0.4573), this result was expected due to its training setup: only synthetic user data and fixed dummy item vectors. The lower recall aligns with its reduced performance on the original data (Table 7), illustrating the tradeoff of excluding item features during shadow model training.
>
> Finally, as requested, we have added the contribution section as Section 1.1 in the revised manuscript.

---

> > ### Comment · Reviewer_domC · 2025-06-09
> >
> > Thank you for the clarification. I have a few further concerns:
> >
> > 1. On point 5 (Gender perturbation (on synthetic data) is more vulnerable than age (based on average score)). According to the table, the average and median scores for gender perturbation of synthetic data are **0.069** and **0.068**, respectively. In contrast, the scores for age perturbation are **0.042** and **0**, respectively. Doesn't this mean that gender is more vulnerable than age?
> >
> > 2. I am still confused by the claim at the beginning of Section 4.2 (Black-Box model exhibits robustness to perturbations in the original dataset). From the result in Sec. 4.1, and as in your reply, that *the original dataset only shows significantly higher scores for age and ZIP code*. Does this indicate the model is not robust to perturbation? Also, what do you mean by "While the high age sensitivity may stem from the perturbation method, our attack accuracy remains unaffected". Can you elaborate a bit more?

---

> > > ### Author Response · Authors · 2025-06-09
> > > **Authors Response**
> > >
> > > Thanks for bringing up further question of yours.
> > >
> > > for 1. Yes, you are right, for synthetic data, gender is more vulnerable than age.  Tables 4 & 5 also present the scores for original data: gender score is 0.063 (average) and 0.065 (median), which are lower than the age scores 0.416 (average) and 0.490 (median). So, for original data, age is more vulnerable than gender.
> > >
> > > But please note that our aim is NOT to say which one, gender or age, is more vulnerable. Instead, we aim to decide which of the two variables is more helpful for data recovery. By Table 4, gender has similar scores based on original data (0.063) and synthetic data (0.070), while Table 5 says that age has more different scores based on original data (0.416) and synthetic data (0.042). For instance, given a new data we can calculate its age score, say 0.35, which is closer to 0.416, and we can say the new data is closer to the original data. By contrast, the gender score is less powerful since the gender scores based on original data (0.063) and synthetic data (0.070) are too similar.
> > >
> > > For 2. Yes, we agree that the higher scores of age indicate less robustness. Please note that the zip code score is small (0.0022 for original and 0.0000 for synthetic; see Table 2), which indicates more robustness.
> > > We agree that our sentence "While the high age sensitivity may stem from the perturbation method, our attack accuracy remains unaffected" is confusing and should be elaborated. As we claimed in response to your first comment, we care more about the high age sensitivity, which may enhance attack accuracy. So this sentence essentially means "While the high age sensitivity may stem from the perturbation method, our attack accuracy can be improved."

---

### Review · Reviewer_h5ok · 2025-07-03

**Summary Of Contributions:**

The work presents an agnostic label-only MIA for the two-tower neural-network models wherein the model outputs continuous values instead of categorical ones. In particular, for recommendation systems, the model outputs are continuous rather than categorical, which motivates the study of such attacks. The authors choose two-tower neural-network models due to their high prevalence in the recommendation systems space. The model outputs per-user and per-item vector scores and pairwise (user-item) relevance scores are often generated using cosine similarity or dot products. The proposed ALOA attacks do not require probability output scores or the original data distribution's knowledge.

First, synthetic data of users and items is created by leveraging the knowledge regarding the schema of the input features. Then models are trained on such data to generate multiple kinds of models that ideally mimic the original model we want to attack (to a certain extent). Since the solution focuses on recommendation systems, the authors prescribe perturbing the original dataset with valid perturbations on user features like gender, ID, zip code, etc, to generate perturbed datasets. Now, the models generated above are used to generate features on perturbed datasets (for data used in training said models) and to assign membership labels. Eventually, the MIA model is developed using input features and membership labels.

Overall, the work uses synthetic data based on input schema, discretized perturbations and attack-specific models to generate the ALOA attacks. Experiments evaluate the expected outcome.

**Audience:**

Yes

**Claims And Evidence:**

Yes

**Requested Changes:**

Answering the above three questions is crucial to understanding the utility of the results.

**Strengths And Weaknesses:**

#### Strengths
The paper is well-presented and demonstrates how the ALOA attacks can be developed.

Empirical results demonstrate the efficacy of the presented mechanism.

#### Weaknesses
Although the authors make the crucial claim that the newly developed ALOA attacks do not require input data distribution, it seems entirely accurate. Due to the direct knowledge of the input schema and bounds on the data, we could assume that we have partial imperfect knowledge of the input data distribution.

Furthermore, the presented work will only work for discrete systems where both user and input features are limited in their cardinality. It's unclear whether the system works for continuous input features. It would be nice to have this distinction highlighted properly. Also, can the authors claim to have a successful attack when the input space is discrete?

The authors mention that the attack's performance heavily depends on the diversity of the synthetic data generated. Can the work properly quantify what diversity exactly implies here? And in what case will the attack fully fail?

The experiments presented are quite minimal, and it would be nice to evaluate on a wide variety of real-life datasets. Furthermore, as previously discussed, mentioning the meaning of dataset diversity and presenting trickier datasets that may not succeed here would be nice for understanding the limitations of the work.

---

> ### Author Response · Authors · 2025-07-15
> **Author Responses**
>
> Thank you for your thoughtful and constructive review. We would like to address your key questions as follows:
>
> 1. On the use of input schema and bounds versus data distribution:
>
> We acknowledge your point and agree that having access to the input schema and bounds provides partial information about the data distribution. However, in real-world scenarios, many commercial models openly specify input constraints (e.g., accepted value ranges, data formats), especially when exposed via APIs. Our approach reflects this realistic threat model: an adversary without access to training data or output probabilities but with the ability to probe the model’s input interface. This setup remains consistent with the agnostic nature of our ALOA attack and aligns with prior assumptions in black-box settings.
>
> 2. Applicability to discrete vs. continuous input spaces:
>
> We agree that the current experiments are tailored to systems with discrete user features, such as occupation or bucketized age. Our objective in this paper is to demonstrate that label-only membership inference attacks can succeed even without classifier outputs, specifically on models that operate in continuous embedding spaces—a departure from prior MIA literature.
> That said, we believe the attack would generalize to continuous inputs. In fact, it may even benefit from continuous inputs since perturbations can be more granular, allowing for finer analysis of robustness in local neighborhoods of the feature space. We plan to explore this direction in future work.
>
> 3. Clarifying the role of synthetic data diversity and failure cases:
>
> As noted in the paper (Section 4.4), the diversity of the synthetic data significantly affects the performance of the shadow model—and by extension, the attack classifier. This diversity is operationalized in our study by contrasting two approaches:
> A dummy approach using a single fixed vector for item features
> A combined approach with fully randomized synthetic item inputs
> Our results show a marked performance difference between the two, demonstrating the role diversity plays in mimicking the embedding distribution of the original training set.
>
> As for failure cases: while not the primary focus of this paper, we recognize their importance. A likely failure scenario occurs when the perturbation strategy is poorly aligned with the model’s sensitivity. For instance, if all perturbed features have negligible influence on the model’s output (e.g., ZIP code in our case), the resulting robustness scores would not differ meaningfully between members and non-members—leading the attack model to approximate random guessing (i.e., 50% accuracy).
>
> 4. On broader evaluation:
>
> We appreciate your suggestion regarding evaluation across more diverse datasets. Our current experiments are designed to establish the method's feasibility and provide strong initial evidence. A more comprehensive benchmark analysis across various architectures and datasets is a valuable future direction that we intend to pursue.

---

### Decision · Action_Editor_tcWq · 2025-08-19

**Recommendation:** Reject

**Additional Comments:**

Some additional comments that I thought of while reading the final revision:

- I cannot find any matches for your first reference "Olivier Chapelle, Qi Ni, and Jim McAuley. Two-tower neural networks for large-scale recommendation systems.". The URL does not point to any relevent site either. Authors need to fix this reference in the future revisions, and carefully check where this seemingly non-exisiting reference is obtained from.
    * Using this reference you write: "Chapelle et al. (2011) has demonstrated that training a single shadow model to mimic the entire output space can achieve performance comparable to training separate shadow models for each category.". Is this just a broken citation? Otherwise the claim sounds unbelievable to me, as this would predate any MIA works by several years!

- For future revisions of the work, I would strongly encourage the authors to clarify the experimental setting.
    * In MIA, you typically try to test whether a particular sample $x_{\text{target}}$ was a member of the $D_{\text{train}}$ that was used to train the target model $m_{\text{target}}$.
    * As far as I understand, your Alg. 2 applies perturbations to the original data before the shadow model training and MIA evaluation.
    * It is therefore not clear, if the success in MIA is due to these perturbations or is the target model actually vulnerable by itself.
    * One could argue that the high vulnerability for the perturbed samples demonstrates that the two-tower models can memorize some samples. However, showing MIA vulnerability on _real_ samples is typically expected from a attack paper.
    * Additionally, in many MIA works the vulnerability is measured in terms of TPR at fixed (low) FPR. This allows to better understand the trade-offs the MIA classifier does in predicting the membership of the samples. While the precision and recall communicate something very similar (recall being the TPR), reporting the results in more common metrics would better position the results in the field of MIA.

- You suggest that the proposed approach falls into the "data distribution agnostic" branch of the MIA, I don't know if this is completely the case. It seem to me that the shadow data sets are constructed such that the shadow model predictions match the target models behaviour on the targeted samples. While I think this approach makes sense, I think it inherently uses some distributional information of the original model through the prediction matching. I would suggest considering this for the future revisions of the paper.

**Audience:**

Yes

**Audience Explanation:**

All the reviewers, as well as I, agree that the studied problem is interesting. Hence I believe there would be interest for this paper among the TMLR audience. That said, the limited amount of evidence undermine the attractiveness of the findings.

**Claims And Evidence:**

No

**Claims Explanation:**

The paper presents a solution for data distribution agnostic membership inference attacks applied against two-tower neural network (TTNN) recommendation systems. The aim is to demonstrate that agnostic label only attacks (ALOA) can be extended for this particular model.

While authors do present evidence that the ALOA can be succesfully applied on TTNN systems, I think in its current form it falls a bit short. Specifically considering the evaluation criteria:
**Accuracy**: While the evaluation metrics seem to match what is often done in the MIA literature, I think the empirical evaluation pipeline has some unclear parts that would require further explanation, including whether the evaluation is based on the actual original data and original "target model". It seems that in many parts of the algorithm, authors apply some perturbations to the data, and I'm a bit worried whether this affects the vulnerability in a way that would not manifest in real-world application.
**Convincing**: The method is applied only for a single data set (and a single model?). In order to make the claims more convincing, more results on real-world data sets would be needed. Otherwise it is hard to say whether the approach actually works, or if it is only a phenomenon of this particular data set. The lack of more comprehensive evaluation was also brought up by a reviewer in their recommendation.
**Clarity**: While authors did improve the clarity after the reviews, I think there is still a lot in the setting that could be clarified. Authors could try to contrast their approach to the existing works to better explain the choices made in the paper (e.g. synthetic data generation, training of the shadow models, attack evaluation.). The clarity issues was also pointed out by some reviewers in the recommendation.

All in all, I think the paper presents an interesting study, but the approach would need further empirical evidence in order to be accepted.

**Resubmission Of Major Revision:**

The authors may consider submitting a major revision at a later time.